# Challenges and Research Priorities for Dementia Care in Malaysia from the Perspective of Health and Allied Health Professionals

**DOI:** 10.3390/ijerph182111010

**Published:** 2021-10-20

**Authors:** Roshaslina Rosli, Michaela Goodson, Maw Pin Tan, Devi Mohan, Daniel Reidpath, Pascal Allotey, Shahrul Kamaruzzaman, Ai-Vyrn Chin, Louise Robinson

**Affiliations:** 1Ageing and Age-Associated Disorder Research Group, University of Malaya, Kuala Lumpur 50603, Malaysia; mptan@ummc.edu.my (M.P.T.); shahrulk@um.edu.my (S.K.); avchin@um.edu.my (A.-V.C.); 2Research Department, Newcastle University Medicine Malaysia, Gelang Patah 79200, Malaysia; Michaela.Goodson@newcastle.edu.my; 3Global Public Health, Jeffrey Cheah School of Medicine and Health Sciences, Monash University Malaysia, Subang Jaya 47500, Malaysia; devi.mohan@monash.edu (D.M.); daniel.reidpath@monash.edu (D.R.); 4United Nations University International Institute for Global Health, Cheras 56000, Malaysia; unu.iigh.director@unu.edu; 5Population Health Sciences Institute, Campus of Ageing and Vitality Newcastle University, Newcastle upon Tyne NE1 7RU, UK; a.l.robinson@newcastle.ac.uk

**Keywords:** data triangulation, dementia care, challenge, focus group, research priorities, thematic analysis

## Abstract

Few studies to date have evaluated dementia care in Malaysia, and the focus of studies has primarily been on epidemiological and laboratory research. In this study, we aimed to identify potential challenges for the delivery of dementia care in Malaysia and priorities for research and enhancing existing dementia care. This study used thematic analysis to evaluate the open and focus group workshop discussions guided by semi-structured questions. Triangulation of the collected data (sticky notes, collated field notes, and transcripts of discussions) was achieved through stakeholder consensus agreement during a workshop held in 2017. Five main themes as priorities for dementia care were identified: (1) availability of a valued multi-disciplinary care service, (2) accessibility of training to provide awareness, (3) the functionality of the governance in establishing regulation and policy to empower care services, (4) perceived availability and accessibility of research data, and (5) influence of cultural uniqueness. The findings of this study seek to enhance existing dementia care in Malaysia but have potential application for other low and middle-income countries with a similar social and health care set up. The constructed relationship between themes also tries to tackle the challenges in a more efficient and effective manner, as none of these aforementioned issues are standalone challenges. In addition, we demonstrated how a carefully constructed workshop with defined aims and objectives can provide a useful analysis tool to evaluate health and social care challenges in a multidisciplinary forum.

## 1. Introduction

Dementia comprises a collection of symptoms and signs that includes multiple disturbances of higher cortical function [1] interfering with an individual’s ability to perform daily activities. The cognitive domains affected include memory, reasoning, orientation, comprehension, and calculation. Dementia is the leading chronic disorder and contributor to disability, dependency, and need for care among older people worldwide. To improve quality of life in dementia sufferers, effective dementia care is of paramount importance.

Dementia is becoming a global emergency of epidemic proportions, affecting millions of people worldwide. In 2019, there were an estimated over 50 million people with dementia (PwD) globally. By 2050, 131.5 million people will be living with dementia worldwide, with 68% of those individuals situated in low and middle-income countries [2]. Unfortunately, in these countries, healthcare systems are often disjointed and fail to provide sufficient quality of care for the population that they are designed to serve [3].

Malaysia is a middle-income country located in Southeast Asia. In the second quarter of 2021, individuals aged 65 years and above comprised 7.1% (2.31 million) of the 32.66 million Malaysian population [4]. This is estimated to increase to 14.5% by 2040, with increases in median age from 26.3 in 2010 to 38.3 in 2040 [5,6]. A recent population-based survey conducted by the Institute of Public Health among 3774 older persons aged 60 years and over in Malaysia estimated the prevalence of probable dementia to be 8.5% (95% CI: 6.97, 10.22) [7]. As the number of people reaching old age is increasing, there are greater requirements for care provision and the societal burden of dementia, and dependency on informal carers is likely to become an issue [8]. In achieving effective dementia care, it is important to address the potential challenges that are currently faced in delivering dementia care as well as priority areas for research and training.

As with many middle-income countries, few studies to date have evaluated dementia care in Malaysia specifically, and the focus of currently available studies is mainly on comparisons in quality of life of PwD in nursing homes versus home care [9], social support for caregivers [10], caregiver experience [11], caregiver burden [12], and cost of care [13]. Interestingly, one review from 2011 revealed the importance of acknowledging dementia management issues and improving the services for patients. The review also suggested where the efforts should be made by the government and private sectors to promote healthy aging in Malaysia [14].

Although global dementia issues and challenges have been explored previously, further work is needed for areas where there are limited studies and a smaller evidence base. 

The aim of this study was to report, using thematic analysis, the findings of discussions regarding dementia care undertaken by a range of stakeholders involved in dementia care. The research methods employed facilitated debate between experts and stakeholders in a group context on how to prioritize service development, care provision, and research. Qualitative research using focus groups is widely used in academic research to examine perceptions, attitudes, feelings, experiences, and reactions in a way that would not be possible with one-to-one interviews, observations, or questionnaires [15,16]. The methods used in this study were selected to enable better understanding of the interactions between stakeholders and allow stakeholders to visualize the challenges of dementia care from different viewpoints, as hypothetically, a multidisciplinary systematic and integrated approach to improving dementia care is needed to address existing challenges to care in Malaysia.

## 2. Materials and Methods

The National Institute for Health Research—Global Health Dementia Prevention and Enhanced Care (DePEC) program brought together dementia care researchers from a number of countries aiming to evaluate dementia care in three middle-income countries, namely India, Malaysia, and Tanzania. The Malaysian team of researchers comprised representatives from University of Malaya, Monash University Malaysia, United Nations University International Instiute of Global Health, and Newcastle University Medicine Malaysia.

A Malaysian National Institute for Health Research—Global Health DePEC workshop was held at the United Nations University, Kuala Lumpur on 7–9 November 2017. The aim of this workshop was to bring together stakeholders involved in dementia research in Malaysia to generate consensus agreement on priorities for research on dementia care in Malaysia and negotiate the challenges to their delivery.

### 2.1. Participants

Participants comprised representatives from the academic sector, non-governmental organizations, clinical practice, and policy-making bodies identified through purposive sampling and snowballing strategies. This approach ensured that participants had specific knowledge of dementia care in Malaysia. The identified participants were formally invited to attend the workshop and followed up through email and social media messaging.

Permission to publish workshop data anonymously was obtained by informing the participants before the workshop. Their verbal consent was audiotaped at the start of the session.

### 2.2. Design

This qualitative study, used a thematic analysis of the open and focus group discussions conducted at the workshop, guided by semi-structured questions and triangulation of the collected data (sticky notes, collated field notes, and transcripts of discussions) during a workshop.

### 2.3. Data Collection

The flow of the workshop was summarized in Figure 1. Prior to the workshop, potential stakeholders in dementia care were identified by the Malaysian research team, and a consensus agreement on who to invite was undertaken to ensure all stakeholder groups were fairly represented. A discussion was then held on the aims and objectives of the workshop and then the program structure. 

During the workshop, participants were introduced to Global Health DePEC and the aims of the session explained, namely to identify challenges to dementia care in Malaysia and priorities for improving dementia care management. Following the introduction, all participants participated in an open discussion to identify key topics, which were then discussed in smaller focus groups to encourage wider participation and greater stakeholder engagement.

#### 2.3.1. Open Discussion

An open discussion was first held, where all participants were individually invited to address to the wider group. This aimed to identify the potential challenges for the delivery of dementia care and priorities for research and enhancing existing dementia care in Malaysia.

The session was chaired by a researcher from Newcastle, where the notes were written on the whiteboard to facilitate discussion. The documentation process was performed by two rapporteurs for field notes and the discussion was audiotaped.

A discussion was then conducted among the panel of experts, which was representative from public health, government, non-governmental organisations (NGO), academia, and the Ministry of Health (MoH) to identify five research priorities for research and enhancing existing dementia care topics for the focus group discussions.

All experts provided their thoughts on the challenges in turns, these initial thoughts were then grouped into themes on a white board. Those themes were then selected, and the topics identified were: (1) prevention and screening, (2) human resource and training, (3) awareness and health literacy, (4) integrated care and regulation/governance of care, and (5) government strategy and policy journey.

#### 2.3.2. Focus Group Discussion

During the workshop, five research priority areas were identified (Figure 1). These became the topics for the focus group discussions. The five separate focus groups all had a similar group of stakeholders, a note taker, and chairperson to feed back the group consensus. Stakeholders in each focus group included participants from public health, government, NGO, academia, and MoH. The session was facilitated by a member of the research team and assisted by a rapporteur, and field notes and discussions were audiotaped. The semi-structured questions were: (1) What needs to be addressed to improve dementia management?; (2) Who is needed to “make it happen”?; and (3) What is your proposed strategy?. Each of the groups then contributed notes to a sticky board and presented them to the other groups. Field notes were recorded by two rapporteurs, and discussions were audiotaped.

### 2.4. Data Analysis

Audiotapes were transcribed, and thematic analyses were conducted using a methodological triangulation of the sticky notes, collated field notes, and transcripts of discussions. The methodological triangulation of the collected data was conducted to eliminate biases and increase the reliability and validity of the analysis, as well as to ensure a comprehensive understanding of the participants’ views [17]. 

The initial coding of focus group themes was developed by an independent investigator, where words and phrases from each of the sticky notes, collated field notes, and transcripts of discussions were identified. The relationships between themes/sub-themes were also evaluated. 

Reliability of the codes was ensured through discussion with a second investigator. The two investigators met periodically to review, discuss, and confirm the codes in order to understand key ideas and summarize perspectives shared by participants. Disagreements between researchers were resolved through discussion, until a consensus on the content was met, and an explanation of each theme was reached.

## 3. Results

Forty-four experts participated in the one-day workshop (Table 1).

### 3.1. Atmosphere of the Groups

The atmosphere of the workshop was very interactive, informative and stakeholders were mutually supportive. The open group discussion was dominated by experts, so the focus groups enabled other stakeholders to express their views in a less intimidating way. While the majority of the participants were more comfortable being active listeners and mostly agreed with the opinion or information given, the focus groups did allow stakeholders who had been quiet during larger group discussion to feel comfortable expressing their views. The moderators running the groups ensured that all participants had sufficient opportunity to communicate their views and be heard in a supportive environment. 

In the focus group, participants actively listened, provided information, and talked in turn. The body language of most participants was calm, particularly when group sizes were smaller and there was greater confidence to express personal views. The discussion session was interactive and filled with constructive debate. The moderator did not express any views during the session; they only explained the objective of the session and ensured that everyone had a fair opportunity to talk to ensure an inclusive and supported environment for voicing opinions. 

### 3.2. Findings

Analysis of the triangulation of data and thematic analysis revealed five main themes that participants felt could potentially improve the delivery of dementia care in Malaysia. These included (1) availability of valued multi-disciplinary care services, (2) accessibility of training to provide awareness, (3) the functionality of the governance in establishing regulation and policy to empower the care services, (4) perceived availability of research data, and (5) influence of cultural uniqueness. Table 2 presents the themes and sub-themes of the determined potential challenges in delivering dementia care. 

The quotations provided for each of the themes were presented anonymously. As the perspective is from a multi-disciplinary group, views are broad and represent a number of viewpoints from health and social care experts, carers, NGOs, and government representatives.

A.Themes and Sub-themes

#### 3.2.1. Availability of a Valued Multi-Disciplinary Care Service

##### Access to Care Services

In general, it was felt that the availability of dementia care provisions and regulated specialized dementia/Alzheimer’s care services in Malaysia was very encouraging, as some service providers offered specialized care and often had set up their own websites or even produced an app to facilitate access for recipients to stay connected with their service provider. However, our findings showed access to care services could be seen as a challenge outside urban environments when there is inadequate information on how to access care services, including support for carers and respite care.


*“There was a need to improve the information for dementia patients and carers on access to dementia care including, prevention strategies, diagnosis, prognosis, treatments, access to respite care, support for caring including formal paid carers and informal unpaid carers and access to “regulated specialist care facilities.”*



*“Quality of care where geographical inequalities in East Coast resulting in a poor distribution of services.”*


Furthermore, the need for a clear dementia pathway to guide healthcare providers in hospitals were emphasized. The existing dementia care pathway was felt to be unclear, and not all services were available everywhere. Many stakeholders felt it was a major obstacle to healthcare providers providing effective management.


*“Clear pathway at the hospital, so that healthcare providers know what to do when presented with a patient with dementia, lack of training and confidence in managing a person with dementia.”*


There was a suggestion that, to ensure effective and standardized management for people with dementia, referral to a number of services supporting dementia patients should be made compulsory at diagnosis.

#### Providing Care Services and Integration between Multi-Professional Working

As a care service provider, some felt that they did not have the skills to deal with all people involved in the informal care of people with dementia, as expectations between patients, family members, and extended family could be very different. They also felt that expectations differed between older and younger generations and those who had lived in other countries where dementia care pathways might be more defined. 

In general, stakeholders felt that, for carers, the care service industry was certainly not an easy field to work in. To be an effective caregiver, it was felt that some training or a specific standardized qualification was important. Unfortunately, the inadequate workforce to meet the demand for care meant that not all carers had been trained or had qualifications, and they did not necessarily want to be identified as carers, as it was not always perceived as a glamourous or significant occupation. Consequently, family and volunteers without specific training provide a significant level of informal care in Malaysia


*“Caring is not seen as a popular occupation and as such is provided mainly through family members and NGOs.”*


Subsequently, the care service providers may need to consider importing non-local caregivers. However, the origin of the carers also raised some concerns among the participants.


*“Where do they (carer) come from (local)?”*


Another related problem highlighted in the discussion is the suitable ratio of caregiver to PwD, as well as the cost and financial burden. The total direct and indirect costs that need to be considered by family members are often underestimated.

The integration in multi-professional working in providing care is seen as an important key point that needs to be explored to enhance dementia care. This multi-professional group comprises medical professionals from primary care, including general practitioners, psychiatrist, geriatrician, neurologists, general physicians, and medical students; individuals of allied health, including nurses, physiotherapists, occupational therapists, dentists, speech therapists, dietitians, and optometrists; and others from the Ministry of Health, NGOs, senior citizen clubs, corporate social responsibilities (CSR) of private companies, health practitioners, and traders or the business sector.

However, the reported challenges in achieving an efficient integration between these professionals working together included lack of coordination, collaboration, and communication between healthcare provision and intra-ministry.


*“Advocate more on cluster systems in hospital—in collaborating services, not all hospital in states are practicing cluster system.”*



*“There are services available in primary, secondary, tertiary healthcare and institute of gerontology—but all are working in silos.”*



*“Malaysia culture for example in rural areas there are Penghulu (head of the village), head of the mosques (and other religion) we should also need to integrate with them in certain ways.”*


It was felt that people might need to be incentivized to encourage them to take up caring roles and a creative or ‘outside the box’ strategy might be needed. 


*“How to make it sexy?”*


Ensuring a healthy work practice was also thought to be important in encouraging sustainability of existing caregivers and attracting interested personnel. Respite care options for care givers were also identified as a necessary priority area


*“Respite and days out (exchange ideas and de-stress). Balance required.”*


To achieve a good integration between multi-professional working, conducting grand ward rounds between hospital departments is potentially beneficial.


*“Collaborating between departments in the hospital through grand ward round.”*


#### Conducting Assessment for Dementia

Cognitive assessment for dementia is a measurement of individual abilities in performing daily life activities that require a higher mental function. The availability of reliable tools is questioned mainly because the tools were not necessarily adapted or validated for the Malaysian language, activities of daily living in Malaysia, or education level attainment of older people in Malaysia. 


*“We also need a good cognitive diagnostic or screening tool.”*


Furthermore, as the severity of dementia progresses, proxy information is needed. However, the validity of proxy information to be used in assessing for dementia has become a concern.


*“Usually the people with dementia are very severe before they present with behavioral problems. Caregivers often come to the memory service as a proxy as participants cannot attend due to disability.”*


One priority area to manage existing dementia was to regularly follow up and evaluate patients using validated tools across the country. The tools needed to be easy to use by non-specialists and validated for use in the main Malaysian languages.


*“Standardized the tools for screening at clinics and the Malaysian Medical Council.”*


Organizing a screening campaign and focusing on a strategy for early detection (by including those <60 years old) was also suggested. A budget allocation for incentives for those who contribute to dementia health and social care work was seen as a potential incentive for this. 


*“Incentives for those who take up geriatric/psychogeriatric to do dementia care. Incentive in screening the “carrot approach” among the healthcare workers.”*


#### Support for Care Services

Obtaining funding for the care service industry, especially from the government, is a challenge due to outdated healthcare policies and funding (1990) for dementia care. Obtaining funding support from the private sector is also difficult. 


*“Economic burden of dementia in Malaysia—presented idea or concept to Ministry of Finance—what is the return in investment? Return on Investment matter needs to be taken care of.”*


The inadequacy in support for social care provision and inefficiency of inter-ministerial collaboration between home help service and welfare department to look out for those who need help are also seen as challenges.


*“We also lack social care provision. Feedback received is that such service is only applicable for those who are receiving welfare aid otherwise they are yet to be eligible for the service. We need to fast-track to assists these patients who are alone at home. Otherwise, family members need to quit their jobs to look after family members with dementia without carer support and become depressed. Collaborate with Welfare Department and (provide) info(data) for the Professors, we do have “Pusat Aktiviti Warga Emas” (Community Activity Centre for the Elderly), we can collaborate in terms of day-care activity for a person with dementia which is yet to be available.”*


There are several suggested priority areas identified to enhance the support systems for existing dementia care. This currently involves a number of ministries, including the Ministry of Housing and Local Government (concerning housing issues for elderly), Ministry of Women, Family and Community Development (concerning the welfare of older persons), and Ministry of Health (concerning health issues), together with the participation of the general public. Furthermore, having an established carer support unit, focusing on psychological and emotional support is certainly important to ensure a strong support system. 

A potential suggested solution concerning the funding is for the government or CSR to provide funding for free training programs available for both informal and formal caregivers.

#### Recognition of Carer Credibility

Lack of recognition of carers’ credibility was also seen as a challenge in ensuring a valued care service. The participants reported that there was no recognized qualifications for carers, no training in many cases, and it is often not recognized as an occupation, rather an activity in addition to another job.


*“There are qualifications or licensing required for people to act as a carer and consequently no government (recognized) diplomas to qualify for such a role and in many cases no training.”*


#### 3.2.2. Accessibility of Training to Raise Awareness

##### Awareness on the Availability

Despite the fact that general information on dementia is widely available throughout various media platforms, participants reported on the need for improvement for information dissemination and accessibility of dementia. Furthermore, the lack of awareness on the accessibility of dementia policies was also highlighted. 


*“We (patients, carers and NGO’s representative) felt that there was a need to improve the information for dementia patients and carers on access to dementia care including, prevention strategies, diagnosis, prognosis, treatments.”*



*“Lack of awareness on policies, and what is (healthcare provision) available in Malaysia?”*


The suggested priority area in enhancing dementia care is certainly through promoting awareness, using all media platforms to spread the message.


*“*
*Make everyone talk about it. Messages need to be catchy, sexy, and viralled to be spread around.”*



*“Make (Create an) epidemic on ‘awareness on issues regarding ageing and related disease’.”*


#### Availability of Training for Multi-Disciplinary Working

Attending a training program is important for those working closely with PwD. This is to ensure they are certified with good skills, updated knowledge on what is available, and effectively managing PwD. However, the lack of training available for multi-disciplinary stakeholders was commonly reported. The lack of specialty training and exposure, especially among healthcare and allied healthcare providers, was felt to be a key factor in preventing people taking up careers as carers or working in the dementia sector. 


*“There are specific training pathways for some doctors, but there is a lack of training for other health professionals and carers. Lack of training and confidence in managing a person with dementia.”*


The challenge in setting up the training is on the most effective methods of training to be used (center-based vs. online). Furthermore, the suitable length of the training course and certification level are being questioned by the Ministry of Health and academic stakeholders.


*“Certification level (certificate/diploma)? Duration of training (weeks/months)?”*


Concerning the priorities in enhancing existing dementia care, participants highlighted the importance of early training. They also suggest conducting exchange training programs with other countries to learn from other countries or set benchmarks on the best approach.


*“*
*Take other countries example as a benchmark. Review their policies and best approach. Japan for example. (We) Can do (an) exchange training program.”*


It was felt that, for health-related training programs, the Ministry of Health and Universities would be “the main organiser”, but the training syllabus should include multi-disciplinary working and involve non-medically-trained participants. Revising training modules and organizing refresher courses for this multi-disciplinary work was also said to be necessary:


*“Identified “who” needed to be trained.”*



*“*
*For allied health, aged care specific training (not only dementia focused), nationwide multi-disciplinary training (combining core training for the first semester on basic principles of aged care).”*



*“*
*Training module available from primary healthcare providers for doctors, staffs, nurses, community nurse, however, it was launched many years ago. But perhaps needs refreshment course for the staffs.”*


#### Exposure on Dementia

The adequacy of dementia education is being questioned when there is a lack of exposure to issues related to dementia, especially among medical officers being reported.


*“Medical officers are not trained and not exposed to these issues (dementia, elderly care…).”*


The suggested priorities in enhancing the existing dementia care include establishing an early education and life-long learning program and ensuring exposure related to dementia issues present at all levels and for all.


*“Education and training should start from young.”*



*“Focal points/committees under the council covering long-life learning programmed, health, etc. Ministry of Higher Education—long-life learning”*



*“In medical, exposure needs to start from every level, postgraduate exposure, interim while waiting for posting. There is no such thing as “too much exposure.” Exposure should be present at all levels and need to be exposed as early as possible.”*


#### 3.2.3. The Functionality of the Government in Establishing Regulation and Policy to Empower the Care Services

##### Need for Dementia Care Policies

The absence of specific policies for dementia was highlighted when the need for dementia care policies being reported. Participants were aware of the available National Policies for Older Persons; however, it was outdated. 


*“*
*There is no specific dementia care policy. Policies and funding for caring for patients with dementia in general as many are out of date from the 1990s. Where the National Policies for Older Persons in 1996 were revised in 2011 and National Health Policy for Older Persons developed in 2008.”*


Furthermore, the process of the implementation of the available policies and action plans for older persons is questionable. It is not clear whether any of the policies were regularly evaluated and audits undertaken on policy directives.


*“Is there any coordination and mechanism toward these (available) policies? Is there any coordination mechanism? Council? Who are the members?”*


There are suggested priorities in enhancing existing dementia care, including reviewing other countries’ policies and action plans and using them as a reference to review and update our own. There is also a suggestion for a policy by the government that encourages job opportunities after retirement.


*“*
*Ministry of Human Resource—the older person still requires to work to earn living.”*


#### Development of Guideline and Model of Dementia Care

The reported challenge with our model of dementia care is the unsustainable potential of caregivers, because the current model relies heavily on filial piety, which is not sustainable, as people need to work and take care of young family members as well as the elderly.


*“*
*Appropriate models for dementia care in middle-income countries as the current model relies heavily on filial piety which is not sustainable given that many young people now wish to live overseas.”*


Furthermore, the adaptation or validation in terms of cost-effectiveness and recommendations of Western dementia care guidelines was felt to be necessary to ensure the guidelines were relevant to the Malaysian setting.


*“*
*Limitations of Dementia care guidelines from Western countries where the cost-effectiveness of recommendations may be irrelevant for Malaysia.”*


The credibility of the members of the council governing guidelines and policies were also questioned on whether the coordination and mechanism of effort towards the development of dementia care policies were present.

The participants highlighted a priority in enhancing dementia care by making regulation according to a standard guideline.


*“Standard protocol to follow—for the tools utilized.”*


#### Regulation and Governance for Care Services

The potential issues of unlicensed home care are indeed worrying, especially safety, abuse, and financial exploitation. However, the participants reported a reluctancy for all unlicensed care homes to be forced to close down their care services.


*“*
*Many unlicensed homes operating but with the new bill, they will be shut down. “Inadequate care” is better than “no care”.”*


The importance of governance on the quality of care and individual practitioners’ regulation was also reported. 


*“*
*Governance on the quality of care and regulating individual practitioners isn’t currently possible with general Malaysian Medical Council requirements. There are qualifications or licensing required for people to act as a carer and consequently no governance diplomas to qualify for such a role and in many cases no training.”*


There was no suggestion on the priority concerning this theme.

#### 3.2.4. Perceived Availability of Research Data

##### Actual and Perceived Lack of Evidence

It was felt that there may be research evidence or census data supporting various aspects of care, but this was not openly available for analysis or integration with other data, for example, population projections or cost benefit analyses.


*“*
*Lack of research in general on the evidence base for various ethnic groups in Malaysia.”*



*“Studies have been subjective and poor quality. The quality of research conducted is also poor.”*



*“*
*Few prevalence studies of dementia and cognitive impairment have been conducted in Malaysia and most have been undertaken in small non-representative samples. We also lack accurate prevalence rates.”*


However, the actual geographical inequalities of epidemiological studies, since studies focus on the West coast of Malaysia, was also reported.


*“*
*East coast Malaysia has less epidemiology than West Coast and modeled statistics for the country may be inaccurate.”*


The suggested priorities to be the focus in research on enhancing dementia care are the need of accurate epidemiology statistics for urban and rural areas and data for all states, not just for large urban conurbations. Furthermore, there was felt to be a need for more accurate data on actual incidence and prevalence rates of dementia in Malaysia, as many felt this was not available or had not yet been collected. 


*“Accurate epidemiology statistics for both rural and urban setting.”*



*“No research has been conducted on incidence.”*


#### 3.2.5. Influence of Cultural Uniqueness

##### Stigmatization towards PwD

The stigmatization towards people living with dementia is strongly influenced by cultural views. These most commonly affect close family members, carers, and the community.


*“Issues on stigma amongst family, carers, and community.”*


#### Customization of Nature of Reaching Out

The influence of cultural uniqueness contributes to challenges in providing dementia care. The nature of reaching out and method of interaction reported may need to be customized according to cultural differences and geographical area.


*“Nature of reaching out and interaction/needs: Focusing on rural & urban differently.”*


#### Practicality of Filial Piety Concepts

The influence of cultural uniqueness contributes to challenges in providing dementia care, especially on the practicality of filial piety concepts.


*“*
*Appropriate models for dementia care in middle-income countries as the current model relies heavily on filial piety which is not sustainable given that many young people now wish to live overseas.”*


There are no suggestions on the priority concerning this theme.

B. Constructed Themes Analysis

In this section, the flow of the discussion was referred accordingly to the relationship of constructed themes and sub-themes in Figure 2 and is discussed further in discussion chapter. The relationship between themes revealed interesting and unexpected findings that highlight multiple of priorities area that can and need to be tackled together to enhance the existing dementia care in Malaysia.

The availability of a valued multi-disciplinary care service and sub-themes such as inadequacy information on how to “access for care service (A1)” were found to be connected with sub-themes of the “awareness on the availability (B1)” and the “actual and perceived lack of evidence (D1).” The “awareness on the availability (B1)” of the care services and resources and the “actual and perceived lack of evidence (D1)” were connected with the inadequacy information on how to “access for care service (A1).” Stigmatization (E1) was connected with the lack of “exposure to dementia (B3)”, and the lack of “awareness in the availability (B1).” The challenge in “providing care services and integration between multi-professional working (A2)” was found to be connected with the needs of “customization on nature of reaching out (E2).” The challenges in “conducting assessment for dementia (A3)” were directly connected with “development of guideline and Model of Dementia care (C2)”. The need for “development of guideline and Model of Dementia Care (C2)” was also found to be connected with the “practicality of filial piety concepts (E3)” in the model of care. The lack of “support for care services (A4)” connected with the urgent “need for Dementia Care Policies (C1)”. Finally, the relationship was found between the “recognition on carers credibility (A5)” challenge in ensuring a valued dementia care service with an absence of strong “regulation and governance for care services’ (C3)”.

## 4. Discussion

This study identified the potential challenges for the delivery of dementia care and priorities for research and enhancing existing dementia care in Malaysia. Findings from this thematic analysis and triangulation of data revealed five main themes that potentially challenge the delivery of dementia care here, which are the availability of valued multi-disciplinary care service, accessibility of training to provide awareness, functionality of governance in establishing regulations and policies to empower the care services, perceived availability of research data, and influence of cultural uniqueness. 

There are, clearly, a number of challenges that can potentially impact upon the delivery of dementia care. When considering the availability of a valued multi-disciplinary care service, sub-themes such as inadequacy information on how to “access for care service” were found to be connected with sub-themes of the “awareness on the availability” and the “actual and perceived lack of evidence.” The availability of dementia care provision and regulated specialized dementia/Alzheimer’s care service in Malaysia was found to be very encouraging. In terms of dementia care provision, family members and PwD can easily access the memory clinic services. This study found that the geographical inequalities (rural vs. urban) in the distribution of services significantly contribute to a challenge. This is supported by a previous study, in which family members of PwD felt that access to care service was limited in rural areas, and a prescient barrier was the distance to the available services [18].

Regarding regulated care services and resources in Malaysia, a previous study reported that various community health and support services are available to assist older adults with dementia and their caregivers. The Social Welfare Department of Malaysia does provide health care, guidance, counseling, recreation, religious teaching, and welfare services, however. In this study, the NGOs were found to be actively involved in the welfare and support provision for the elderly, often trying to coordinate services themselves. The community support services in palliative home cares and daycare centres were already set up in large cities but less so outside of Selangor and Kuala Lumpur [14]. The interesting relationship between sub-themes of “awareness on the availability” of the care services and resources and the “actual and perceived lack of evidence” with the inadequacy information on how to “access for care service” in this study revealed that people did not really know how to access a service, even if they knew it existed, and sometimes, they did not know services were available. This is a little disheartening, as it implies there may be an underutilization of some excellent facilities [2,19]. A study has reported that around 60% of the general public think that there are inadequate community services in place for people living with dementia and for carers, but if health care providers think similarly, this is an obvious early area to tackle [2].

If we look at the patent of the constructed relationship of the themes, it is revealed that these are the consequence of the challenge of “stigmatization towards PwD”. Stigmatization was connected and caused by the lack of “exposure to dementia (B3)”, especially among family members, caregivers, and the community, and leads to the lack of “awareness in the availability” of the information regarding dementia care. Our finding was validated by the most recent 2019 World Alzheimer Report by Alzheimer’s Disease International, with almost 70,000 respondents from 155 countries. Stigmatization is still found to be a major barrier to people seeking out information, help, advice, support, and even a diagnosis, as well as preventing or delaying people from putting plans in place [2].

The challenge in “providing care services and integration between multi-professional working” was found to be connected with the need for culturally relevant and specific offerings in specific geographical areas. This relationship revealed the necessity of having an appropriate approach to the development of care services depending on the local demographic and needs, particularly when dealing with the “difficult to reach” elderly. In this study, it was felt that there was underdiagnosis in those living in rural areas, and these people would not, therefore, be able to access specialized dementia care [20].

In this study, the challenges in “conducting assessments for dementia” were highlighted, as the absence of standardized tools was directly connected to the “development of guidelines and Models of Dementia care”. Some people felt that a guideline with one standard tool to be used by all healthcare providers would solve the issues with diagnosis. However, one size does not fit all, because diagnosing dementia is often difficult due to differences in language, culture, literacy rates, and other socio-economic and demographic factors [21]. There are a number of validated tools in Malaysia, such as the IDEA cognitive screen and visual-based cognitive assessment tools, available to serve our uniqueness [22,23].

From another angle, the need for “development of guidelines and a Model of Dementia Care” was also found to be connected with the “practicality of filial piety concepts” in the model of care. This study interestingly revealed that the most practiced model of dementia care, which relies heavily on filial piety, is not sustainable, especially for those who are situated in the urban area. The filial piety concepts were no longer practical/suitable for application among today’s generation. This may be because children are mostly attached to an obligation of commitments and job responsibilities to earn money for a living. In contrast, the sense of obligation was reported in previous studies in 2018 and 2003, where both authors of the studies suggested that filial obligation/caregiving role acceptance may be the primary motive for caregiving as a result of embedded religious values [11]. Especially in Chinese traditional culture, the value of filial piety in Chinese society is great [10].

One interesting point that contributes to this area of research is the relationship between the challenging of lack of “support for care services”, especially in outdated health policies, as well as funding (1990) for dementia care and the urgent “need for Dementia Care Policies”. This relationship revealed that this is the first qualitative study that emphasizes the urgent need for specific dementia care policies in Malaysia, as no prior research was found echoing this assertion. We believe this is due to one of the strong and unique characteristics of this study, where the participants comprised of groups of experts, stakeholder, and NGOs that have a direct connection with dementia care. Having a specific policy is seen as a potential solution for funding support, where it will gain the stakeholder’s interest to invest in dementia care sectors, as they will be aware of the awaiting mutual benefits.

Finally, a relationship was found between the “recognition of carers credibility” and the challenge in ensuring a valued dementia care service with the absence of strong “regulation and governance for care services”. Interestingly, this relationship itself suggested a potential solution for the recognition challenges, whereby having the regulations that govern the certification of qualifications among formal and informal caregivers will potentially enhance the existing quality of care.

The use of mixed open discussions and a focus group methodology in this workshop allowed access to multidisciplinary stakeholders and data presenting a deeper understanding of the challenges and priorities for dementia care based on individual knowledge and experiences. One limitation of this study, however, is that the workshop and the participants were recruited using purposive sampling and snowballing strategies, but we believe that this approach to be valid as considerable effort was put into triangulating perceptions and data. The sampling procedure approach used here ensured that a large number of stakeholders involved in dementia care was represented, and consequently, we believe the views expressed here to be novel data, and there was a genuine desire for interdisciplinary work to improve dementia care in Malaysia.

## 5. Conclusions

This study demonstrates how dementia care quality can be enhanced in the coming years in Malaysia. The findings of this study are also likely to be applicable to other low- and middle-income countries with a similar health and social care set up. Having identified priority themes for health service management and research, stakeholders can focus their efforts into integrated multidisciplinary working. 

This methodology of study is also a novel way to present qualitative data gained at workshops to stimulate research priority areas. 

The limitation of this study is the small sample size. However, the representative opinions expressed by the participants are equally valid, because they are from the health and allied health professional in multidisciplinary areas. The open large group discussion and focus group discussion methods also enabled in-depth discussions, where the saturation of information from the collected data was achieved.

The future direction of this research is to focus on the urgent needs of dementia care policy development. There is ongoing research on policy development and how stakeholders can influence dementia policies in LMICs.

## Figures and Tables

**Figure 1 ijerph-18-11010-f001:**
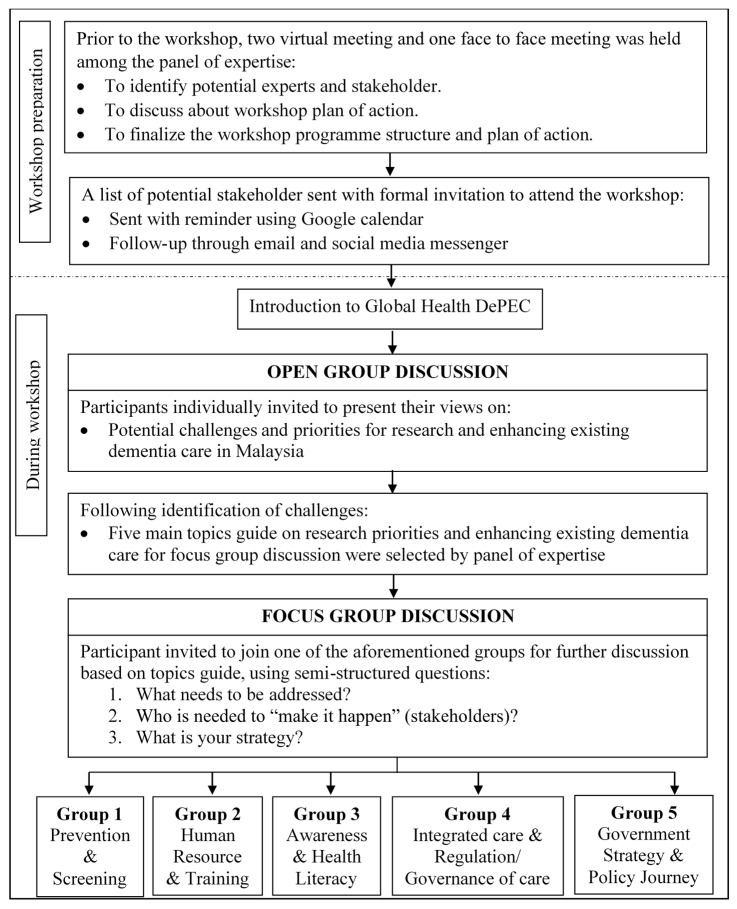
The flow of Malaysian DePEC Workshop. Note: DePEC = dementia prevention and enhanced care.

**Figure 2 ijerph-18-11010-f002:**
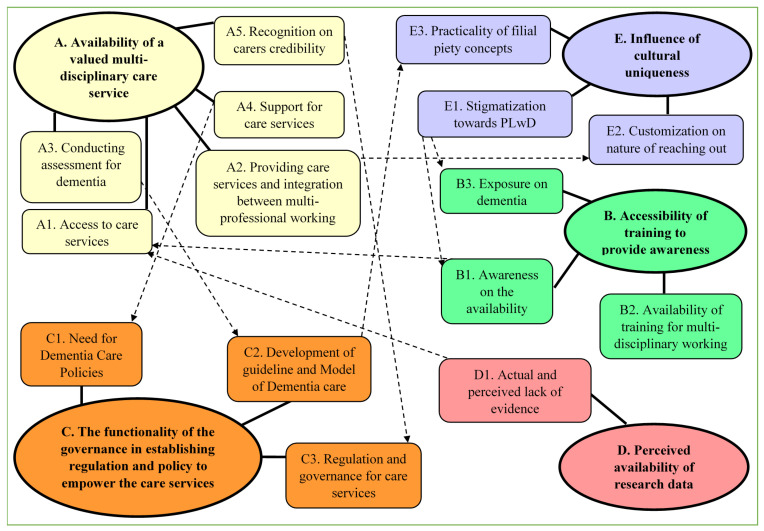
Constructed themes relationship.

**Table 1 ijerph-18-11010-t001:** Demographic characteristic of participants (*n* = 44).

Characteristics	*n* (%)
**Gender**	
Male	14 (31.8)
Female	30 (68.2)
**Background**	
Medicine	18 (40.9)
Psychiatry/Psychology	4 (9.1)
Policymaker/Government	2 (4.5)
United Nations	4 (9.1)
Non-Governmental Organization/Social workers	5 (11.4)
Demographers/Methods	1 (2.3)
Public Health/Epidemiology	6 (13.6)
Neuroscience	1 (2.3)
Gerontology	3 (6.8)

**Table 2 ijerph-18-11010-t002:** Five themes on potential challenges for the delivery of dementia care and priorities for research and enhancing existing dementia care.

Themes	Sub-Themes
A. Availability of a valued multi-disciplinary care service	1. Access to care services
	2. Providing care services and integration between multi-professional working
	3. Conducting assessment for dementia
	4. Support for care services
	5. Recognition of carers’ credibility
B. Accessibility of training to provide awareness	1. Awareness of availability
	2. Availability of training for multi-disciplinary working
	3. Exposure to dementia
C. The functionality of the governance in establishing regulation and policy to empower the care services	1. Need for dementia care policies
	2. Development of guidelines and model of dementia care
	3. Regulation and governance for care services
D. Perceived availability of research data	1. Actual and perceived lack of evidence
E. Influence of cultural uniqueness	1. Stigmatization of PwD
	2. Customization on nature of reaching out
	3. Practicality of filial piety concepts

Notes: PwD = People with dementia.

## Data Availability

Not applicable.

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
