# Peer review of "Challenges and Research Priorities for Dementia Care in Malaysia from the Perspective of Health and Allied Health Professionals"

_ijerph, 2021, doi:10.3390/ijerph182111010_

Round 1
Reviewer 1 Report
I’m sorry, but the article is still difficult to read due to the level of English. Someone needs to read it through thoroughly and sort this out. I’ve given a few examples in my specific comments below, but I’m not proofreading the whole article for you.
I didn’t read the whole article as it felt like I would be so distracted by the language that I wouldn’t be fair to the actual content of the article. From the little I did read, I appreciate that you have made changes, but many of them do not make sense or need rewriting.
Specific comments on the bits I did read
Abstract line 23 – should be ‘This study contributes’
Line 32 – should be ‘Dementia comprises a collection of’ (the word ‘a’ was there in the original, so I don’t know why it is removed now)
Line 33 – should be ‘interferes with an individual’s ability to perform daily life activities including’ (I’m fairly sure I gave this as an example in my previous comments, so I’m surprised it hasn’t been resolved)
Lines 40-41 – should be something like ‘In 2019, it was estimated that there were over 50 million people with dementia (PwD) globally.’ At present, the sentence does not make sense.
Lines 42-44 – this sentence is not a sentence in its own right. It is a sub-clause of the previous sentence. It either needs to be incorporated into the previous sentence, or rewritten to make it make sense as a standalone sentence. E.g.
- By 2050, 131.5 million people will be living with dementia worldwide. 68% of them are situated in low and middle-income countries [2] where the healthcare systems are often disjointed and fail to provide sufficient quality of care for the population that they are designed to serve [3].
- By 2050, 131.5 million people will be living with dementia worldwide, where 68% of them are situated in low and middle-income countries [2]. In these countries the healthcare systems are often disjointed and fail to provide sufficient quality of care for the population that they are designed to serve [3].
Line 47 – should be ‘population’ not ‘populations’
Lines 110-114 – needs rewriting as it’s too long and doesn’t make sense.
Lines 115-116 – this is a repetition of lines 98-100, so is unnecessary. I’m also not sure as a reader whether I care if they were sent a Google calendar reminder.
Author Response
Dear Reviewer,
Thank you for being very patience in addressing the extensive need for English language editing. The manuscript now has been revised by my two co-authors, where English is their first language. All of your time and effort are highly appreciated. This enabling us to improve our data presentation and publication.
For the response, please find attached. Thank you.
Best regards,
Roshaslina rosli

Reviewer 2 Report
I believe that most of the comments have been duly answered. Those that have not been answered are because they are impossible to answer at the present time and are due to the deficiencies of the design phase.
While it is true that the article does not contribute new knowledge to the area of dementias, I do consider of interest the publication of work from developing countries and with a multidisciplinary approach. For the latter reason, I do recommend acceptance for publication but I suggest that for future work of similar characteristics, the option of greater participation of family/domestic caregivers and also of new technologies (e.g., telemedicine) should be expanded.
Author Response
Dear Reviewer,
Thank you so much for all of your comments and suggestions. All of your time and effort are highly appreciated. Certainly, its enabling us to improve our data presentation and publication. Thank you.
Best regards,
Roshaslina Rosli
Reviewer 3 Report
Dear Authors,
I enjoyed reading the revised version of your manuscript.
All the best.
Author Response
Dear Reviewer,
Thank you so much for all of your comments and suggestions. All of your time and effort are highly appreciated. Certainly, it is enabling us to improve our data presentation and publication. Thank you.
Best regards,
Roshaslina Rosli
This manuscript is a resubmission of an earlier submission. The following is a list of the peer review reports and author responses from that submission.
Round 1
Reviewer 1 Report
This paper presented a study to identify the challenges for the delivery of dementia care and priorities for research in Malaysia. It made a qualitative study using a mix of an open discussion and focus group discussion. It showed five main themes that potentially challenge the delivery of dementia care.
Overall:
- The paper is organized well with proper structure, and the bibliography is sufficient and well given.
- The presented methodology and the results are communicated. The novel contribution of the paper is highlighted.
This paper can be accepted. However, there are some points that the authors can handle:
- On pages 1 and 5, you wrote “Five main themes that potentially a challenges for the delivery of dementia care was availability of valued multi-disciplinary care service: ‘accessibility of training to provide awareness’, ‘the functionality of the governance in establishing regulation and policy to empower the care services’, ‘perceived availability of research data’ and ‘influence of cultural uniqueness’”. You put four themes I single quotes and I think you should put this also “availability of valued multi-disciplinary care service” in single quotes.
- The abbreviation must be written beside the words when it appears for the first time. So, the authors need to check all of them. Example: DePEC was mentioned without writing the meaning for the first time. Also, “RR and MPT” are mentioned without writing the meaning.
- You have to check the references carefully because you wrote references 5,6 twice at line 51 on page 2.
Reviewer 2 Report
I think this could be a really interesting article, but unfortunately it is difficult to read and follow due to the level of the English used. Many sentences are missing words or do not make sense. I am not going to spell check/grammar check the whole article, but recommend that someone proofreads it closely and thoroughly before resubmission. At present, as a reader I’m distracted by the language and trying to work out what is being said, rather than focusing on the actual content.
For example, the first sentence is currently ‘Dementia comprises a collection symptoms and disturbance of multiple higher cortical functions [1] and interfere individuals ability performing daily life activities includes memory, reasoning, orientation, comprehension and calculation.’
It should be something like: ‘Dementia comprises a collection of symptoms and disturbances of multiple higher cortical functions [1] and interferes with an individual’s ability to perform activities of daily life including memory, reasoning, orientation, comprehension and calculation.’
The remainder of my comments ignore the English used, and try to focus on what I believe is actually being reported.
Specific comments
Abstract (these comments were made before I gave up trying to address the English)
Lines 17-19 – The sentence ‘This is a …a workshop’ needs reworking as it currently doesn’t make sense.
Lines 20-24 – The first part of the sentence ‘Five main… care service:’ doesn’t make sense. It also only lists four themes in ‘’.
Lines 24-27 – The sentence ‘This study… countries’ doesn’t make sense. If the first bit was ‘This study contributes to the enhancement of existing dementia care by identifying five main themes of potential…’ it might be better.
Lines 27-29 – The sentence ‘The constructed… effective’ doesn’t end properly. Should it be something like ‘…in a more efficient and effective manner’?
Line 51 – Why does it have [5-6]. (5, 6).? Remove unnecessary bits.
Line 90 – what is DePEC? Please don’t assume that a reader will know.
Lines 98-99 – why were ethical review and approval waived, and who by?
Lines 122-130 – It seems odd to refer the reader to Figure 1 to see the semi-structured discussion guide, then give the 3 questions used just afterwards. It seems a bit pointless and annoying as I scrolled back to the figure before realising that the questions were given in the next paragraph.
Line 139 – instead of ‘developed by RR’ I would suggest ‘developed by one of the investigators’. You could then have ‘Reliability of the codes was ensured through discussion with a second investigator. The two investigators met periodically to review, discuss and confirm the codes in order to understand the key ideas and summarise the perspectives shared by the participants.’
Lines 149 and 161 – they are both 3.1. Findings should be 3.2, which affects all sub-sections that follow. Please check the numbering of other sections to ensure accuracy.
Line 193 – why do you introduce PwD as an abbreviation, when you’ve already been using PLwD?
Line 203 – why do you reintroduce the abbreviation PLwD when you’ve already been using it?
Line 226 – the abbreviation NGO should be expanded when it is first used in line 212, not here.
Line 276 – ROI needs to be expanded
Pay attention to the use of capital letters. E.g. in line 387 you have Model of Dementia Care but in line 388 you use Model of Dementia care. Also, in line 395 why is it Western Dementia not western dementia?
Line 444 – there is a ] that needs to be removed.
Figure 2 – how were the relationships between themes/sub-themes identified? They are presented without really saying where they came from. If they were part of the discussion between investigators, it should be mentioned there. If not, it needs to be explained where the relationships came from. I know in lines 477-478 it says ‘were found to be connected’ but not how/by whom. I would also suggest that Figure 2 should be part of the results section rather than the discussion section. I appreciate this could mean a lot of work though, as lines 475-555 would need to be looked at to split out bits that explain the findings, and bits that take the discussion further. E.g. lines 476-479, the sentence ‘In the availability…(D1)’ is more of a finding, while lines 479-485 are more of a discussion.
Reviewer 3 Report
First of all, I would like to thank the authors for their work. Any work that aims to make visible the reality of the care situation of patients with dementia is important in itself.
I will now proceed with some suggestions for changes that I believe could help improve the quality of the article:
- At present I consider it a mistake to speak of dementia in a generic way. The research required for non-neurodegenerative dementias is different from that required for degenerative dementias. Within the latter, the research and treatment needs also vary depending on the specific type. A subanalysis according to degenerative or non-degenerative causes and according to specific diagnosis would have made the article much more interesting.
2. In Table 1 I consider it a mistake to separate Medicaina and Psychiatry into 2 distinct categories. I understand that they are not 2 exclusive categories but that psychiatry is a subcategory of medicine. I understand the same for geriatrics and I think that neuroscientist is a neurologist, if so, idem. I think that more weight in the discussion by neurologists and geriatricians would have contributed a lot to the discussion. Social workers, psychologists or neuropsychologists were not considered for the discussion? It is a bit surprising really. The contribution to the discussion of surveys administered to family members and to the patients themselves with a diagnosis of mild cognitive impairment would also have been very interesting. The patient's own perspective is often not taken into account and is the one that best reflects their needs and the difficulties they are experiencing or have experienced.
3. In the section on access to care services, it is surprising that not enough mention is made of the difficulties that may exist with specific diagnostic tools. Currently, diagnostic management has changed radically with respect to cognitive impairment. A diagnosis based on biomarkers increases the specificity of diagnosis and is essential to provide access to clinical trials and in the future (fortunately not too distant) to disease-modifying treatments that will arrive soon.
It is also surprising that no mention is made of the creation of multidisciplinary groups with regular online contact. New technologies allow collaboration between primary care and highly specialized hospital services in a simple way. Online consultation between specialties, validation of screening tests for telematic use and thus homogenizing access to different resources are also essential, aspects that are not discussed in the paper.
Performing an isolated test such as the MMSE screening test is misleading. It would be necessary to consider combining it with a more specific test for memory (the most frequently affected domain). Also, tests that assess the burden of affective and/or other neuropsychiatric symptoms, and functionality, which could be self-administered or easily hetero-administered by low-skilled personnel such as the IDDD and NPI should also be included in a first evaluation.
Information should be added on difficulties of access to established pharmacological treatment. Whether or not there is public funding for access to these drugs. It is also essential to also influence the public funding of cognitive stimulation to be applied in the context of dementia. Unknown but relevant aspects for people who do not know the reality of Malaysia.
4. I don't think that Figure number 4 contributes anything. It is not intuitive at all. It would be necessary to redo it and think about the message to be conveyed with this figure.
In summary, I find the subject matter of the article interesting. I also find the identification of the main challenges acceptable, although with certain discrepancies. The sample size seems to me too small (very little representation of medical specialties involved in the care of these patients; of specialized nursing...), heterogeneous, and without direct representation of the figure of the patient and his or her relatives. But what I find less adequate are the conclusions reached as possible answers to the challenges posed. In general, the answers seem to me to be simple, well known and do not contribute anything new to the field. One would expect concrete initiatives to emerge from these discussions, and this could have been of more interest and could have been extrapolated to other countries in a similar socioeconomic situation.
At the methodological level, I do not find any noteworthy errors, but I must admit that I am not used to this type of qualitative analysis. I believe that a telephone or online survey with all neurologists, geriatricians, psychiatrists, primary care physicians, psychologists, neuropsychologists, social workers, occupational therapists, neurology/psychiatry/geriatrics nurses of the public health system and the participation also of those patients with mild cognitive impairment and/or mild dementia and their relatives would provide much more information than the work presented.
Reviewer 4 Report
Dear Authors,
Your paper deals with a very important issue of global health – both for policy and research.
In the following, please find my comments and recommendations:
Abstract: Please, precise and avoid redundancies.
Introduction: A paragraph on the state of play of dementia care and dementia research in Malaysia is missing. Show the knowledge gap and derive the aim of the paper and the key questions.
Materials and Methods: Is this really a mixed-methods approach? You are triangulating findings from two qualitative research methods.
Please, insert the background of your research at the beginning of this chapter. Then, information on data collection should follow (former chapter 2.3).
Figure 1: Please, explain the contents of this illustration in the text in more detail. There are some questions left open: Who was part of the panel of expertise? How were the “five main topics” selected in the open group discussion? Provide more information on methodology! As you mention in the text 44 experts participated in the focus group discussion. Did each expert only participate in one single group or did each expert participate in all of the five groups?
2.3. Data analysis: Please, describe the chosen method of thematic analysis. Furthermore, a description of the process of data triangulation is needed.
Results:
3.1 Atmosphere of the groups: In my opinion, this paragraph could be shifted to the chapter “Materials and Methods”. If not, please explain the relevance of content of this paragraph. Was there a dominancy of certain experts during the discussion(s)? How did you cope with “opinion-leaders”?
The presentation of the findings is a queue of direct quotations. Please, provide information on the professional background of the person you are citing by using abbreviations, e.g. “….” (A 1) for an expert from the academic field.
In my opinion, the findings contain little information on the “purpose of the paper”, namely challenges of and research priorities for dementia care in Malaysia.
Please, reconsider the title of your paper!
Discussion:
This section contains information on previous studies on the issue in Malaysia. Please, shift them to the chapter “Introduction”. Information on the cited studies is incomplete (e.g. line 486, page 13).
Point out the originality of your research. Discuss the reliability and validity of your findings.
Figure 2 is appropriate for the “Results” chapter.
Discuss the gender dimension of your empirical findings.
Conclusions:
Avoid redundancies and provide conclusions for research and practice (policy and administration) as well as further research directions.
All the best!